# GENERALIZATION ERROR BOUNDS FOR NEURAL NETWORKS WITH ReLU ACTIVATION

## ABSTRACT

We show rigorous bounds on the generalization error for Neural Networks with ReLU activation under the condition that the network size doesn't grow with the training set size. In order to prove these bounds we weaken the notion of uniform stability of a learning algorithm in a probabilistic way by positing the notion of almost sure (a.s.) support stability and proving that if an algorithm has low enough a.s. support stability its generalization error tends to 0 as the training set size increases. Further we show that for Stochastic Gradient Descent to be almost surely support stable we only need the loss function to be locally Lipschitz and locally smooth with probability 1, thereby showing low generalization error with weaker conditions than have been used in the literature. We then show that Neural Networks with ReLU activation and a doubly differentiable loss function possess these properties, thereby proving low generalization error. The caveat is that the size of NN must not grow with the size of the training set. Finally we present experimental evidence to validate our theoretical results.

## 1 INTRODUCTION

Given the importance of the generalization error in machine learning there have been numerous theoretical approaches to bounding it, one of the most significant being the concept of algorithmic stability whose origins can be traced back to Vapnik & Chervonenkis (1974). In this paradigm the definition of uniform stability by Bousquet & Elisseeff (2002) was a landmark because it allowed the authors to prove sharp concentration bounds showing that the generalization error for the algorithms that satisfy this property tends to 0 as the training set size increases. Thanks to the success of Bousquet & Elisseeff (2002) on a variety of non-neural methods, when Hardt et al. (2016) showed that a learning function computed by optimizing certain loss functions through Stochastic Gradient Descent is uniformly stable under mild conditions on the loss function, this clearly raised the expectation that the next step would be to show that Neural Network-based models are also uniformly stable and, hence, have a low generalization error. This analysis was then expected to corroborate the empirically observed fact that the generalization error is low for NN models (c.f., e.g., Krizhevsky et al. (2012), Hinton et al. (2012).) Case closed. However, this triumphal march of algorithmic stability was rudely interrupted by Zhang et al. (2017), who showed explicit examples of distributions that cannot be learned with low generalization error and yet can be learned with an NN of an appropriate size. Zhang et al. (2017) speculated that the problem arose from the fact that the concept of uniform stability was based *only* on properties of the algorithm and not on the properties of the unknown data distribution. While this was a reasonable speculation, it left open some questions: Is there a reasonable definition of stability that incorporates distribution properties and leads to small generalization error? And can we show that NNs with non-linear activation functions such as ReLU at the neurons have this property?

In this paper we give a strongly affirmative answer to the first question and a partially affirmative answer to the second question. We define a notion called *almost sure (a.s.) support stability* which is a probabilistic weakening of uniform stability. Unlike the data-dependent notions of stability defined in Kuzborskij & Lampert (2018), Lei & Ying (2020) that bound generalization error in expectation, a.s. support stability can be used to show high probability bounds on generalization error. This proof entails proving a mild generalization of McDiarmid's Inequality that could be of independent interest. The major contribution of our paper is that we show how to handle the non-linearity of ReLU in a mathematically rigorous way by modifying a result of Milne (2019) to show that ReLU

affects the smoothness of the gradients computed by Stochastic Gradient Descent with probability 0 for well-behaved distributions. To the best of our knowledge such a result has not been reported in the literature.

Clearly the question arises: What about the evidence presented by Zhang et al. (2017) that there exist unknown distributions that can be learned by an NN but will always have unbounded generalization error? To answer this we need to understand that our analysis of SGD presented requires the loss function to be smooth and Lipschitz in the parameter space (though with an appropriate probabilistic weakening of these properties.) Our analysis is able to demonstrate that these properties hold for NNs whose size doesn't grow with the size of the training set. So, our specific answer that we have for the second question listed above: *Almost support stability can be used to demonstrate that the generalization error for NNs with ReLU as long as the size of the NN doesn't grow too fast with the training set size, and if the unknown distribution places probability 0 on sets of Lebesgue measure 0.* This leaves open the matter of a theoretical explanation for why certain heavily overparametrized NNs generalize well in practice. However, we do provide some directions on how our analytical framework may be used to prove such results.

In particular our contributions are:

• In Section 3 we define a new notion of stability called *a.s. support stability* and show in Theorem 2 that algorithms with a.s. support stability $o(1/\log^2 m)$ have generalization error tending to 0 as $m \to \infty$ where $m$ is the size of the training set.

• In Section 4 we show that if we run stochastic gradient descent on a parametrized optimization function that is only *locally* Lipschitz and *locally* smooth in the parameter space and that too only on input points selected with probability 1, then the replace-one error is bounded even if the training is conducted for $c \log m$ epochs. This implies (Corollary 7) that any learning algorithm trained this way has generalization error that goes to 0 as $m \to \infty$ for learning rate $\alpha_0/t$ at step $t$ for an appropriately selected value of $\alpha_0$ that does not depend on $m$.

• In Section 5 we show that the output of an NN with ReLU activations when used with a doubly differentable loss function is locally Lipschitz and locally smooth in the parameter space for all inputs except those from a set of Lebesgue measure 0 (Theorem 9). This allows us to show a.s. support stability and a generalization error bound for NNs with size that doesn't grow with $m$.

• We experimentally verify our theoretical results in Section 6, showing that our bounded Lipschitz and bounded smoothness conditions hold in practice.

## 2 RELATED WORK

Several authors have that although NNs are known to generalize well in practice, many different theoretical approaches have been tried without satifactorily explaining this phenomenon, c.f., Jin et al. (2020); Chatterjee & Zielinski (2022). We refer the reader to the work of Jin et al. (2020) which presents a concise taxonomy of these different theoretical approaches. There are also several works that seek to understand what a good theory of generalization should look like, c.f. Kawaguchi et al. (2017); Chatterjee & Zielinski (2022). Our own work falls within the paradigm that seeks to use notions of algorithmic stability to bound generalization error that began with Vapnik & Chervonenkis (1974) but gathered steam with the publication of the work by Bousquet & Elisseeff (2002).

The applicability of the algorithmic stability paradigm to the study of generalization error in NNs was brought to light by Hardt et al. (2016) who showed that functions optimized via Stochastic Gradient Descent have the property of uniform stability defined by Bousquet & Elisseeff (2002), implying that NNs should also have this property. Subsequently, there was renewed interest in uniform stability and a sequence of papers emerged using improved probabilistic tools to give better generalization bounds for uniformly stable algorithms, e.g., Feldman & Vondrak (2018; 2019a) and Bousquet et al. (2020). Some other works, e.g. Klochkov & Zhivotovskiy (2021), took this line forward by focussing on the relationship of uniform stability with the excess risk. However the work of Zhang et al. (2017) complicated the picture by pointing out examples where the theory suggests the opposite of what happens in practice. This led to two different strands of research. In one thread an attempt was made to either discover those cases where uniform stability, (e.g. Charles & Papailiopoulos (2018)), or to show lower bounds on stability that ensure that uniform stability does not

exists, (e.g. Zhang et al. (2022)). The other strand of research, our work falls in this category, focuses on weakening the notion of uniform stability, specifically by making it data-dependent, thereby following the suggestion made by Zhang et al. (2017). Kuzborskij & Lampert (2018) defined "on-average stability" which is weaker than our definition of a.s. support stability. Consequently, their definition leads to a weaker in-expectation bound on the generalization error where the expectation is over the training set as well as the random choices of the algorithm. Our Theorem 2, on the other hand, provides a sharp concentration bound on the choice of training set. Lei & Ying (2020) define an "on-average model stability" that requires the average replace-one error over all the training points to be bounded in expectation. While their smoothness requirements are less stringent, the problem is that their generalization results are all relative to the optimal choice of the weight vector, which implies a high generalization error in case of early stopping.

## 3  A.S. SUPPORT STABILITY AND GENERALIZATION

In this section we present a weakening of the notion of uniform stability defined by Bousquet & Elisseeff (2002) and show that exponential concentration bounds on the generalization error can be proved for learning algorithms that have this weaker form of stability.

### 3.1  TERMINOLOGY

Let $\mathcal{X}$ and $\mathcal{Y}$ be the input and output spaces respectively. We assume we have a training set $S \in \mathcal{Z}^m$ of size $m$ where each point is chosen independently at random from an unknown distribution $D$ over $\mathcal{Z} \subset \mathcal{X} \times \mathcal{Y}$. For $z = (x, y) \in \mathcal{Z}$ we will use the notation $x_z$ to denote $x$ and $y_z$ to denote $y$. Let $\mathcal{R}$ be the set of all finite strings on some finite alphabet, and let us call the elements of $\mathcal{R}$ *decision strings* and let us assume that there is some probability distribution $D_r$ according to which we will select $r$ randomly from $\mathcal{R}$. Further, let $\mathcal{F}$ be the set of all functions from $\mathcal{X}$ to $\mathcal{Y}$. In machine learning settings we typically compute a map from $\mathcal{Z}^m \times \mathcal{R}$ to $\mathcal{F}$. We will denote the function computed by this map as $A_{S,r}$. Since the choice of $S$ and $r$ are both random, $A_{S,r}$ is effectively a random function and can also be thought of as a randomized algorithm.

Given a bound $M > 0$, we assume that we are given a bounded *loss function* $\ell : \mathcal{Y} \times \mathcal{Y} \to [0, M]$. We define the *risk* of $A_S$ as

$$R(A, S, r) = \mathrm{E}_{z \sim D}\left[\ell(A_{S,r}(x_z), y_z)\right],$$

where the expectation is over the random choice of point $z$ according to data distribution $D$. Note that the risk is a random variable since both $S$ and $r$ are randomly chosen. The *empirical risk* of $A_S$ is defined as

$$R_e(A, S, r) = \frac{1}{|S|} \sum_{z \in S} \ell(A_{S,r}(x_z), y_z).$$

We are interested in bounding the *generalization error*

$$|R(A, S, r) - R_e(A, S, r)|.$$

### 3.2  A WEAKENING OF UNIFORM STABILITY

Given $S = \{Z_1, \ldots, Z_m\}$ chosen randomly according to $D^m$, we choose $\{Z_{1+m}, \ldots, Z_{2m}\}$ also from $D^m$ independently from $S$ and, for each $i \in [m]$, we define

$$S^i = \{Z_1, \ldots, Z_{i-1}, Z_{i+m}, Z_{i+1}, \ldots, Z_m\}.$$

**Definition 1** (Almost (Sure) Support Stability). *We say an algorithm $A_{S,r}$ has $\eta$-almost support stability $\beta$ with respect to the loss function $\ell(\cdot, \cdot)$ if for $Z_1, \ldots, Z_{2m}$ chosen i.i.d. according to an unknown distribution $D$ defined over $\mathcal{Z}$,*

$$\forall i \in [m] : \forall z \in supp\,(D) : E_r\left[|\ell(A_{S,r}(x_z), y_z) - \ell(A_{S^i,r}(x_z), y_z)|\right] \leq \beta$$

*with probability $1 - \eta$. If $\eta = 0$ then we say the algorithm has* almost sure (a.s.) support stability $\beta$.

We note that this notion weakens the notion of uniform stability introduced by Bousquet & Elisseeff (2002) by requiring the bound on the difference in losses to hold with a certain probability. This probability is defined over the random choices of $Z_1, \ldots, Z_{2m}$. Besides the condition on the loss is required to hold only for those data points that lie in the support of $D$. These conditions make almost support stability a *data-dependent quantity* on the lines of the suggestion made by Zhang

et al. (2017). We also observe that almost support stability is comparable to but stronger than the hypothesis stability of Kearns & Ron (1999) as formulated by Bousquet & Elisseeff (2002).

## 3.3 EXPONENTIAL CONVERGENCE OF GENERALIZATION ERROR

A.s. support stability can be used in place of uniform stability in conjunction with the techniques of Feldman & Vondrak (2019a) to give guarantees on generalization error. In particular we can derive the following theorem.

**Theorem 2.** *Let $A_{S,r}$ be a algorithm that is symmetric in distribution and has a.s. stability $\beta$ with respect to the loss function $\ell(\cdot, \cdot)$ such that $0 \leq \ell(A_{S,r}(x_z), y_z) \leq 1$ for all $S \in \mathcal{Z}^m$, for all $r \in \mathcal{R}$ and for all $z = (x_z, y_z) \in \mathcal{Z}$. Then, there is a constant $c > 0$ s.t. for any $m \geq 1$ and $\delta \in (0, 1)$, with probability $1 - \delta$,*

$$E_r \left[ R(S, r) - R_e(S, r) \right] \leq c \left( \beta \log(m) \log \left( \frac{m}{\delta} \right) + \sqrt{\frac{\log(1/\delta)}{m}} \right).$$

*Proof outline.* We give a high-level outline here. Feldman & Vondrak (2019b) used two steps to get a better generalization guarantees. The first step is range reduction, where the range of the loss function is reduced. For this they define a new clipping function in Lemma 3.1 Feldman & Vondrak (2019a) which preserves uniform stability and hence it will also preserve a.s. support stability. They also use uniform stability in Lemma 3.2 Feldman & Vondrak (2019a) where they show the shifted and clipped function will still be stable which is done by applying McDiarmid's inequality to $\beta$ sensitive functions. Here use a modification of McDiarmid's Inequality (Lemma 12 given in Appendix A) to get bounds for a.s. support stability. The second step is dataset size reduction (as described in Section 3.3 Feldman & Vondrak (2019a)) which will remain the same for a.s. support stability as this only involves stating the result for a smaller dataset and the probability, and then taking a union bound. Therefore both steps of the argument given in Feldman & Vondrak (2019a) go through for a.s. support stability.

## 4  A. S. SUPPORT STABILITY OF STOCHASTIC GRADIENT DESCENT

A large family of machine learning algorithms follow a paradigm in which the learned function is parametrized by a vector $\boldsymbol{w} \in \mathbb{R}^n$ for some $n \geq 1$, i.e., we have some fixed function $g : \mathbb{R}^n \times \mathcal{X} \rightarrow \mathcal{Y}$. The training set is used to learn a suitable parameter vector $\boldsymbol{w} \in \mathbb{R}^n$ such that the value $g(\boldsymbol{w}, x_z)$ is a good estimate of $y_z$ for all $z \in \mathcal{Z}$. This value of $\boldsymbol{w}$ is learned by running Stochastic Gradient Descent (SGD) using a training set drawn from the unknown distribution. We will say that the size of the training set is $m$ and the number of steps for which SGD is trained is $T$ with the parameter vector at step $t$ being denoted $\boldsymbol{w}_t$ for $0 \leq t \leq T$. Assuming that $T$ is a multiple of $m$, the algorithm moves in *epochs* of $m$ steps. To frame the learned function output by this algorithm in the terms defined in Section 3.1, the random decision string $r$ consists of the pair $(\boldsymbol{w}_0, (\pi_0, \ldots, \pi_{\frac{T}{m}-1})$, i.e., the random initial parameter vector $\boldsymbol{w}_0$ from which SGD begin and the set of $T/m$ random permutations used in the $T/m$ epochs.

### 4.1  SOME PROPERTIES OF PARAMETERIZED FUNCTION

In Hardt et al. (2016), proving that the learning algorithm derived by SGD is stable requires smoothness and Lipschitz properties of $f$, but *only* for partial derivatives taken on $\mathbb{R}^n$, i.e., on the parameter space. The requirement there is that *every function* in the family of functions $\{f(\cdot, z) : z \in \mathcal{Z}\}$ is smooth and has a bounded Lipschitz constant. Our key insight is that this requirement is stronger than required. All we need is that *the functions induced by the data points that we pick to train SGD have these properties.* We now present some definitions that encapsulate this idea.

**Definition 3** (Almost Locally Parameter Lipschitz functions)**.** *Given a distribution $D$ defined over $\mathcal{Z}$, a parametrized function $f : \mathbb{R}^n \times \mathcal{Z} \rightarrow \mathbb{R}$ is said to be $\beta$-almost Locally $L_l$ Lipschitz w.r.t $D$ for some $\beta \in (0, 1]$ if for any $\boldsymbol{w} \in \mathbb{R}^n$ there exists constants $L_l > 0$ and $\epsilon > 0$ such that, with probability $\beta$ (over the choice of z), for all $\boldsymbol{w}' \in \mathbb{R}^n$, $\|\boldsymbol{w}' - \boldsymbol{w}\| < \epsilon$ implies*

$$|f(\boldsymbol{w}', z) - f(\boldsymbol{w}, z)| \leq L_l \|\boldsymbol{w}' - \boldsymbol{w}\|.$$

*Note that $L_l$ can depend on z. If $\beta = 1$ then we say that $f$ is* almost surely locally $L_l$-parameter Lipschitz *(a.s. $L_l$-LPL for short).*

**Definition 4** (Almost Locally Parameter-Smooth functions). *Given a distribution $D$ defined over $\mathcal{Z}$, a parametrized function $f : \mathbb{R}^n \times \mathcal{Z} \to \mathbb{R}$ is said to be $\gamma$-almost $K_l$-Parameter Smooth w.r.t $D$ for some $\gamma \in (0, 1]$ if for any $\mathbf{w} \in \mathbb{R}^n$ there exist constants $K_l > 0$ and $\epsilon > 0$ such that, with probability $\gamma$ (over the choice of $z$), for all $\mathbf{w}' \in \mathbb{R}^n$, $\|\mathbf{w}' - \mathbf{w}\| < \epsilon$ implies*
$$\|\nabla f(\mathbf{w}', z) - \nabla f(\mathbf{w}, z)\| \le K_l \|\mathbf{w}' - \mathbf{w}\|.$$
*Note that $K_l$ can depend on $z$. If $\gamma = 1$ then we say that $f$ is* almost surely locally $K_l$-parameter smooth *(*a.s. $K_l$-LPS *for short).*

If the function (or its gradient) is locally bounded, and, if we only look at this function at a finite number of points, we get a "global" property within this finite set of points:

**Lemma 5.** *Given $f : \mathbb{R}^n \to \mathbb{R}$ we have that (1) if $f$ is bounded and is locally Lipschitz at a finite set of points $A \subseteq \mathbb{R}^n$, then there is a constant $L$ such that for every pair $\mathbf{w}, \mathbf{w}' \in A$*
$$|f(\mathbf{w}) - f(\mathbf{w}')| \le L\|\mathbf{w} - \mathbf{w}'\|, and$$
*2) if $\nabla f$ is bounded and is locally smooth at a finite set of points $A \subseteq \mathbb{R}^n$, then there is a constant $K$ such that for every pair $\mathbf{w}, \mathbf{w}' \in A$*
$$\|\nabla f(\mathbf{w}) - \nabla f(\mathbf{w}')\| \le K\|\mathbf{w} - \mathbf{w}'\|.$$

The proof is in Appendix B.

## 4.2 A.S. SUPPORT STABILITY OF SGD

We now work towards the a.s. support stability of SGD. First we state a theorem that bounds the replace-one error of SGD upto a certain number of epochs.

**Theorem 6.** *Suppose we are given a labelled data set $\mathcal{Z}$ and probability distribution $D$ defined over it. Suppose we have a parametrized loss function $f : \mathbb{R}^n \times \mathcal{Z} \to \mathbb{R}$ that is a.s. $L_l$-LPL and a.s. $K_l$-LPS w.r.t $D$. Suppose for some $T > 0$ multiple of $m$, for each $i \in [m]$ we run Stochastic Gradient Descent on the $f$ for $T/m$ epochs using a training set $S$ of size $m$ chosen i.i.d. according to $D$ with random choices $r$ and, parallely, with the same set of random choices $r$, on a set $S^i$ wherein the $i$-th data point $z_i$ of $S$ has been replaced by another data point $z_i'$ chosen from $D$ independent of all other random choices and with a decreasing learning rate of $\alpha_t \le \frac{\alpha_0}{t}$. Then, if for all $t > 0$, we denote by $\mathbf{w}_t$ and $\mathbf{w}_t'$ the parameter vectors obtained while training with $S$ and $S^i$ respectively, we have that with probability 1*

$$E_r\left[f(\mathbf{w}_T, z) - f(\mathbf{w}_T', z)\right] \le \frac{L_S L_g}{K_g} F(T/m) \cdot \frac{(m^{\alpha_0 K_g} - 1)}{m}, \tag{1}$$

*where $F(T/m) := (2T/m)^{\alpha_0 K_g} \cdot 2^{T/m}$, and $L_g$ and $K_g$ are the "global" Lipschitz and "global" smoothness constants we get from Lemma 5 and $L_S$ is a data-dependent Lipschitz constant which depends only on points picked according to the unknown distribution $D$.*

*Proof outline.* The proof follows the lines of the argument presented by Hardt et al. (2016) with the difference that we allow for a probabilistic relaxation of the smoothness conditions in line with our definition of a.s. stability. Also, note that we have to account for an expectation over the random string $r$, and that we have been able to extend the argument to multiple epochs. The complete proof of Theorem 6 is in Appendix B.

*Data dependence and $L_S$.* A key feature of the bound presented in equation 1 is that the dependence on the data is expressed through the data-dependent Lipschitz constant $L_S$. This constant depends on the training points and the replacement point $z_i'$ which is also picked from the data distribution. In general we expect that if the unknown distribution $D$ has low variance then $L_S$ will be small. Further, a line of research in the optimization literature has shown that the gradients associated with SGD decay as training proceeds, even for non-convex loss functions, c.f. Section 4 of Bottou et al. (2018). Therefore we can conclude that the a.s. stability bound of equation 1 is closely connected to the data distribution and is likely to be useful for cases where SGD returns a meaningful solution and vacuous for bad cases like the one presented by Zhang et al. (2017).

**Corollary 7.** *For a learning algorithm that is symmetric in distribution and trained as described in the statement of Theorem 6 under the condition that $L_g$ is a constant w.r.t. $m$, there is a constant $c > 0$ that depends on $\alpha_0$ and $K_g$ such that if it is trained for at most $c \log m$ epochs the expectation*

*of the generalization error of the algorithm taken over the random choices of the algorithm decreases as $O\left(m^{-(1-\alpha_0 K_g)} + m^{-1/2}\right)$ with probability at least $1 - 1/m$ over the choice of the training set if $\alpha_0 < 1/(K_g)$.*

*Proof.* It is easy to check from Theorem 6 that with the conditions given in the statement of Corollary 7 the learning algorithm has a.s. support stability $\beta$ where $\beta$ is $o(1/m)$ if $\alpha_0 < 1/(K_g)$. We can therefore apply Theorem 2 with $\delta = 1/m$ to get the result. □

If we compare this corollary with Theorem 4 of Kuzborskij & Lampert (2018), we note that their analysis requires an $\alpha_0$ that decreases with $m$ whereas our analysis allows for a value of $\alpha_0$ that is constant. Importantly, their analysis bounds generalization error only up to the end of a single epoch whereas we can bound the error well beyond that. Kuzborskij & Lampert (2018) also require the Hessian to have a bounded Lipschitz constant, i.e., the third derivative of the loss function has to be bounded. We do not need this constraint.

## 5 NEURAL NETWORKS WITH RELU ACTIVATION

The main result of this section presents the conditions required for low generalization error for Neural Networks with ReLU activation:

**Theorem 8.** *For a neural network with ReLU activation and 1 output neuron, trained on set $S \in D^m$ using SGD for $T$ steps, where $D$ is over $\mathbb{R}^d \times \mathcal{Y}$, such that $\mathcal{Y}$ is countable and for each $y \in \mathcal{Y}$ we get a countable set $\{x \in \mathbb{R}^d : Pr_D\{\mathsf{lab}(x) = y\} > 0\}$, where $\mathsf{lab}(x)$ is label of x. For a doubly differentiable loss function with bounded first and second order derivatives and learning rate $\alpha_t = \frac{\alpha_0}{t}$. If the data points of S and the spectral norms of weight matrices explored by SGD are bounded, and $L_g$ and $K_g$ are the "global" Lipschitz and smoothness constant and $L_S$ is the data-dependent Lipschitz constant then,*

$$E_r\left[|R(S,r) - R_e(S,r)|\right] \leq O\left(\frac{L_S L_g}{K_g}\frac{F(T/m)\log(m)^2}{m^{1-\alpha_0 K_g}} + \sqrt{\frac{\log(m)}{m}}\right),$$

*with probability at least $1 - 1/m$, where $F(T/m) := (2T/m)^{\alpha_0 K_g} \cdot 2^{T/m}$.*

The proof of the theorem is at the end of Section 5.1.

First we will now establish that the theory of a.s. support stability applies to NNs under the conditions specified.

### 5.1 SUPPORT STABILITY OF NEURAL NETWORKS WITH RELU ACTIVATION

The key to showing the support stability of NNs with ReLU is to establish that they are parameter-Lipschitz and locally parameter-smooth. We show that this happens with good probability.

**Theorem 9.** *For every $w \in \mathbb{R}^n$, a doubly differentiable loss function, $\ell : \mathbb{R} \times \mathbb{R} \to \mathbb{R}$, applied to the output of a NN with ReLU activation is locally parameter-Lipschitz and locally parameter-smooth for all $x \in \mathbb{R}^d$ except for a set of measure 0.*

*Proof outline.* The proof of this theorem is based on the argument that for a given $w$ a point of discontinuity exists at a given neuron if the input $x$ lies in the set of solutions to a family of equations, i.e., in a lower dimensional subspace of $\mathbb{R}^d$. This proof is an adaptions of an idea of Milne (2019) and can be found in Appendix C.

Theorem 9 begs the question: How large are these Lipschitz and smoothness constants? We provide some general bounds that can be improved for specific architectures:

**Proposition 10.** *Suppose we have a fully connected NN with ReLU activation at the inner nodes that has depth $H + 1$. Then, if the spectral norms of weight matrices are bounded for every layer i.e. $\|W^i\|_\sigma \leq s_i, \forall i \in [H]$, and size of each layers be $\{d_0, \ldots, d_H\}$ then,*

$$L_g \leq \sqrt{\sum_{i=1}^{H}\left((d_{i-1} \cdot d_i)\frac{\prod_{l=1}^{H}\|W^l\|_\sigma}{\|W^i\|_\sigma}\right)} \times \|\boldsymbol{x}\|_2$$

The proof of the proposition is in Appendix C. The bound on $L_g$ should be compared to the bounds given in the context of Rademacher complexity by Bartlett et al. (2017) and Golowich et al. (2018). Our bound is related to the spectral complexity and can potentially be independent of the size of the network. We are now ready to prove our main theorem.

*Proof of Theorem 8.* Theorem 9 tells us that a NN with ReLU activations is locally parameter-Lipschitz and locally parameter-smooth. From Proposition 10 we see that the boundedness of the first and second derivatives of the loss function and the boundedness of the spectral norm of weight matrices and data points ensures that the Lipschitz constants and smoothness constants associated with the NN's training are bounded w.r.t. $m$. With all these in place we can apply Theorem 6 to get the a.s. support stability followed by Theorem 2 to get the desired result. The bound on the smoothness constant also tells us that the constraint on $\alpha_0$ needed to apply Corollary 7 is not too stringent, i.e., $\alpha_0$ can be chosen to be a value that is a constant w.r.t. $m$. □

## 5.2 DISCUSSION

Several of the constraints we have placed above are only for ease of exposition. We discuss which of these constraints can be removed.

• *Removing the fully connectedness constraint.* Although Theorem 8 is stated for a fully connected network, it can be applied to networks like CNNs which have partially connected convolutional layers with intermediate pooling and normalization layers (e.g., LeNet, AlexNet, etc.). In such cases the symmetry in distribution condition required for Theorem 2 holds as long as the training set is chosen i.i.d. from the unknown distribution.

• *Adding regularization terms to the loss function.* Several popular regularizers, the $\ell_2$ regularizer being a prominent example, are doubly differentiable and therefore Theorem 8 can be applied when such regularizers are used along with a doubly differentiable loss function.

• *Activation functions apart from ReLU.* We present a comprehensive treatment of ReLU activation but that does not mean that our results are restricted to this kind of activation. Non-linearities like max-pool can also be handled in our framework by proving that, like with ReLU, the points of discontinuity of such a non-linearity also lie in a set of Lebesgue measure 0.

• *The case of multiple outputs* Although we state the Theorem 8 for the case of a NN with a single output, it is not difficult to extend the technique to cover the case of multiple outputs.

*What about overparametrization?* The question put by Zhang et al. (2017) as rearticulated by various authors, including Chatterjee & Zielinski (2022), can be put as follows: How do we speak theoretically about generalization in the overparametrized regime where NNs often generalize well even though they have enough parameters to overfit? Chatterjee & Zielinski (2022) make the point that some property of the training set is at play here, a claim more specific than the general claim that there is some property of the data distribution that has been missed. These are related, of course, but it is our feeling that a more fine-grained analysis of the growth in the replace-one error is required. We feel that it is possible to look at the local Lipschitz and local smoothness constants in greater detail taking both network structure and input point values in mind. We hope that the polynomial characterization of the NN presented in Section C.1.2 will help this process. We conjecture that it may be able to show that for certain distributions the constants actually improve (decrease) as the training proceeds, and this could lead to a proof of a.s. support stability in these cases.

## 6 EXPERIMENTS

In this section we will experimentally show that the Parameter Lipschitz and Parameter Smoothness constants that we reasoned with are indeed bounded, and that the theoretical upper bound that we derived for the generalization error of a neural network holds in practice.

**Setup.** For our experiments we use *MNIST* and *FashionMNIST* datasets. In both datasets, we randomly selected $20,000$ as training and $1,000$ as test points. All experiments were conducted using a fully connected feed forward neural network with single hidden layer and ReLU activation. We train the model using SGD (batch size = 1), with cross-entropy loss, starting with randomly

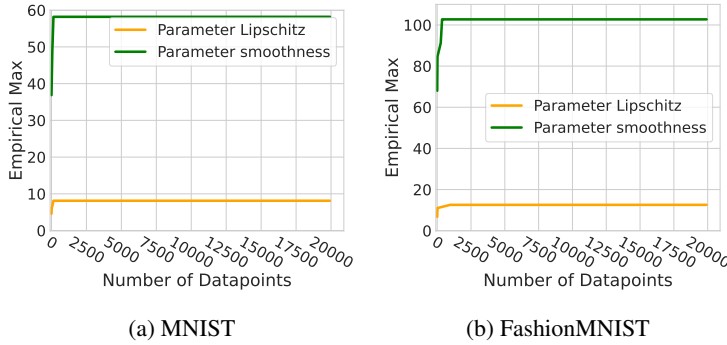

(a) MNIST

(b) FashionMNIST

Figure 1: Maximum of the Lipschitz and smoothness constants at every $p (= 20)$ interval of updates of SGD (we plot both the running average and the highest value found so far). Notice that these constants have a clear upper bound throughout the training process.

initialized weights. As suggested in our analysis we use a decreasing learning rate $\alpha_t = \frac{\alpha_0}{t}$. In each epoch we consider a random permutation of the training set. $L_g$ and $K_g$ are computed by calculating the norm of gradients and Hessian across the training steps and taking their max.

**Experiment 1.** Our first experiment is aimed towards establishing that the "global" Lipschitz ($L_g$) and smoothness ($K_g$) values estimated using local values at each step are bounded. Figure 1 summarizes the results of these experiments over MNIST and FashionMNIST datasets ($\alpha_0 = 0.001$). The plots contain the maximum of the local parameter Lipschitz and smoothness values obtained after running each experiment 10 times with random weight initialization. These results support our Theorem 9, since the upper bound values quickly stabilize and do not grow with the size of training set in both datasets. Similarly, the bounded smoothness constant supports our constraint on the learning rate, $\alpha_0 \le \frac{1}{4K_g}$. We find $L_g$ to be $8.1174$ (MNIST) & $12.5737$ (FashionMNIST), and $K_g$ to be $58.185$ (MNIST) and $102.7096$ (FashionMNIST).

**Experiment 2.** We now turn our attention to the experiment to support our main result, i.e., the empirical generalization error estimated using validation set is upper bounded by our theoretical upper bound. We first split each dataset in 20:1 ratio into training and validation set, and train the model at varying size of training set. We empirically compute the generalization error at each training set size using the validation set. Figure 2 compares this empirical generalization error (in red) with the theoretical upper bound (in blue)From these results, we can see that our bound decreases along with the generalization error thus empirically validating our reasoning.

## 7 CONCLUSION

We have devised a theoretical framework in this paper for using algorithmic stability to prove generalization bounds for NNs with ReLU nonlinearities. We feel that it is possible to prove stronger results in this framework than the ones we have presented here, and more widely applicable ones. Immediate lines of research that suggest themselves are to apply our methods for CNNs and to investigate what other architectures can be approached with our method. It would be particularly

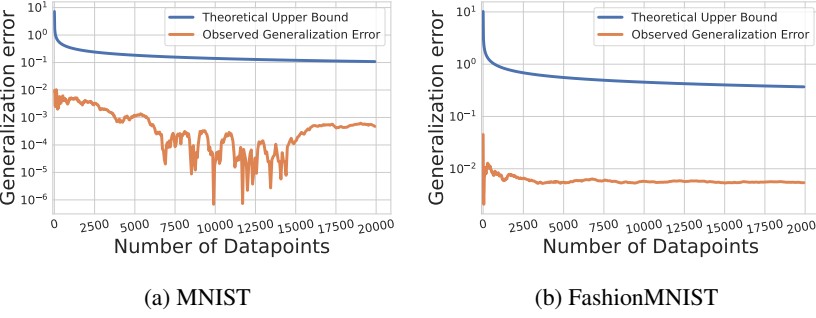

(a) MNIST

(b) FashionMNIST

Figure 2: Comparison of empirical generalization error (red) vs. theoretical upper bound (blue) with varying training set size for different datasets.

interesting to see if there is some analog of our polynomial characterisation for GNNs. Another clear challenge is the overparametrized case. We have not been able to tackle it but we feel a more fine-grained analysis in our framework may possibly lead to such a result.

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

## A  MODIFICATION OF MCDIARMID'S THEOREM

We first define a probabilistic weakening of Lipschitz functions.

**Definition 11.** *Given $2m$ i.i.d. random variables $X_1, \ldots, X_{2m}$ drawn from some domain $\mathcal{Z}$ according to some probability distribution $D$, for some $\beta > 0$ and $\eta \in [0, 1]$, a function $f : \mathcal{Z}^m \to \mathbb{R}$ is called $\eta$-almost $\beta$-Lipschitz w.r.t. $D$ if*

$$\forall i \in \{1, \cdots, m\} : |f(X_1, \ldots, X_m) - f(X_1, \ldots, X_{i-1}, X_i', X_{i+1}, \ldots, X_m)| \leq \beta,$$

*with probability at least $1 - \eta$. In case $\eta = 0$ we say that $f$ is almost surely $\beta$-Lipschitz w.r.t $D$. When $D$ is understood we will omit it, and for the case $\eta = 1$ we will simply write that $f$ is almost surely (or just a.s.) $\beta$-Lipschitz.*

We now state a modified version of McDiarmid's theorem that holds for $\eta$-almost $\beta$-Lipschitz functions.

**Lemma 12.** *Let $X_1, \ldots, X_m$ be i.i.d. random variables. If $f$ is $\eta$-almost $\beta$-Lipschitz and takes values between $0$ and $M$, then,*

$$Pr\{f(X_1, \ldots, X_m) - E[f(X_1, \ldots, X_m)] \geq \epsilon\} \leq \exp\left[\frac{-2\epsilon^2}{m\left((1-\eta)\beta + M\eta\right)^2}\right] + \eta.$$

For simplicity, we adopt the notation $A_{i,j} = A_i, \cdots, A_j$ (i.e. with subscript $(i,j)$ we represent $j - i + 1$ variables) and $A_i$ represents only one variable in the rest of the section. Before we prove Theorem 12, we state and prove the following lemma:

**Lemma 13.** *Let $f : \mathcal{Z}^m \to \mathbb{R}$ be a function such that $0 \leq f(\cdot) \leq M$. Define $V_i = E_{i+1,m}[f(X_{1,m})] - E_{i,m}[f(X_{1,m})]$, where $E_{i,j}[\cdot]$ denotes $E_{X_i, \cdots, X_j}[\cdot]$. Then, if $f$ is $\eta$-almost $\beta$-Lipschitz, then with probability $1 - \eta$, $\forall i, V_i \leq (1-\eta)\beta + \eta M$.*

*Proof.*

$$V_i = E_{i+1,m}[f(X_{1,m})] - E_{i,m}[f(X_{1,m})]$$
$$= \int \cdots \int_{y_{i+1}\cdots y_m} f(X_{1,i}, y_{i+1,m})d\mu(y_{i+1,m}) - \int \cdots \int_{y_i \cdots y_m} f(X_{1,i-1}, y_{i,m})d\mu(y_{i,m})$$
$$= \int \cdots \int_{y_{i+1}\cdots y_m} \left[f(X_{1,i}, y_{i+1,m})d\mu(y_{i+1,m}) - \int_{y_i} f(X_{1,i-1}, y_{i,m})d\mu(y_{i,m})d\mu(y_i)\right]d\mu(y_{i+1,m})$$

where $\mu$ is the measure associated with the random variables. Thus, $V_i$, a function of $X_{1,i}$, is an expectation over the random variables $y_{i+1,m}$. Thus, we can write $V_i = E_{y_{i+1,m}}[\Phi(X_{1,i}, y_{i,m})|X_{1,i}]$.

Now, let's define two sets as follows:
$$S_i = \{x \in \mathcal{Z}^i : \exists y \in \mathcal{Z}^{m-i} : \Phi(x, y) \leq \beta\},$$
$$S'_x = \{y \in \mathcal{Z}^{m-i} : \Phi(x, y) \leq \beta\}.$$

Then, since $f$ is $\eta$-almost $\beta$-Lipschitz, and renaming $y_i$ as $y_{i+m}$ in $\Phi$, since they are *i.i.d.*, for any $x \in S_i$, we have

$$E[\Phi(x, y)|x] \leq \sum_{y \in S'_x} Pr\{y\}\beta + \left(1 - \sum_{y \in S'_x} Pr\{y\}\right)M, \tag{2}$$

and for any $x \notin S_i$, we have
$$E[\Phi(x, y)|x] \leq M.$$

We further note that $\sum_{x \in S_i} Pr\{x\} \geq 1 - \eta$, and $\sum_{y \in S'_x} Pr\{y\} \geq 1 - \eta$. Therefore, applying these bounds to Equation 2, we get, with probability at least $1 - \eta$,
$$V_i \leq (1-\eta)\beta + \eta M.$$

$\square$

*Proof of Theorem 12.* We will use Azuma's inequality to prove the result. First we show that the sequence $V_1, \cdots, V_m$ is a martingale difference sequence with respect to $X_{1,m}$. To see this, we first notice that $V_i$ is a function of $X_{1,i}$. Also, $E[|V_i|]$ is finite since $f$ is bounded. Finally,
$$E[V_{i+1}|X_{1,i}] = E_{i+1,m}[E_{i+1,m}[f(X_{1,m})] - E_{i,m}[f(X_{1,m})]|X_{1,i}]$$
$$= 0$$

Let $\boldsymbol{E}$ be the event that $|f(X_{1,m}) - E[f(X_{1,m})]| \geq \epsilon$, and $\boldsymbol{F}$ the event that $V_i$ is bounded by $\beta(1 - \eta) + \eta M$. We have,
$$Pr\{\boldsymbol{E}\} \leq Pr\{\boldsymbol{E}|\boldsymbol{F}\}Pr\{\boldsymbol{F}\} + Pr\{\overline{\boldsymbol{F}}\}. \tag{3}$$
Using Lemma 13, we have that with probability $1 - \eta$, for all $i$, $V_i$ is bounded by $\beta(1 - \eta) + \eta M$.

Applying Azuma's inequality to $Pr\{\boldsymbol{E}|\boldsymbol{F}\}$, we have the result. $\square$

## B  A.S. Support Stability of SGD Proved

*Proof of Lemma 5.* If $f$ is locally Lipschitz at $\boldsymbol{w} \in A$, there is an $\varepsilon_{\boldsymbol{w}} > 0$ and an $L_{\boldsymbol{w}} > 0$ such that for all $\boldsymbol{w}' \in \mathbb{R}^n$ with $\|\boldsymbol{w} - \boldsymbol{w}'\| \leq \varepsilon_{\boldsymbol{w}}$, $|f(\boldsymbol{w}) - f(\boldsymbol{w}')| \leq L_{\boldsymbol{w}}\|\boldsymbol{w} - \boldsymbol{w}'\|$. So, let us turn our attention

to those $\boldsymbol{w}' \in A$ that lie outside the ball of radius $\varepsilon_{\boldsymbol{w}}$ around $\boldsymbol{w}$. Note that for such a $\boldsymbol{w}'$, if $B > 0$ is the bound on $f$, we have that

$$\frac{|f(\boldsymbol{w}) - f(\boldsymbol{w}')|}{\|\boldsymbol{w} - \boldsymbol{w}'\|} \leq \frac{2B}{\varepsilon_{\boldsymbol{w}}}.$$

Therefore the "global" Lipschitz constant for $f$ within $A$ is $\max\{L_{\boldsymbol{w}}, 2B/\varepsilon_{\boldsymbol{w}} : \boldsymbol{w} \in A\}$ which is bounded since $A$ is finite. This proves the first part of the lemma. The second part follows similarly. $\qquad\square$

*Proof of Theorem 6.* For some $i \in [m]$ we couple the trajectory of SGD on $S$ and $S^i$ where $z_i \in S$ has been replaced with $z_i'$. Our random string $r$ in this case is a random choice of an initial parameter vector, $\boldsymbol{w}_0$, and a random set of $T/m$ i.i.d permutations $\pi_0, \ldots, \pi_{T/m-1}$ of $[m]$ chosen uniformly at random. We use these random choices for training both the algorithms with $S$ and $S^i$. For $0 \leq j \leq T/m - 1$, we denote $\pi_j^{-1}(i)$ by $I_j$, i.e., $I_j$ is the (random) position where the $i$th training point is encountered in the $j$th training epoch. The key quantity we will track through the coupled training process will be

$$\delta_t = \|\boldsymbol{w}_t - \boldsymbol{w}_t'\|,$$

for $1 \leq t \leq T$. If we can show that $\mathrm{E}_r [\delta_T]$ is bounded by some quantity $B$ almost surely, we can invoke the fact that $f$ is a.s. $L_l$-LPL to say that $\|\mathrm{E}_r [f(\boldsymbol{w}_t, z) - f(\boldsymbol{w}_t', z)]\| \leq L_g B$ for all $z \in \mathrm{supp}(D)$, where $L_g$ is the "global" Lipschitz constant we get from Lemma 5, which is defined for all $z$.

We argue differently for the first epoch and differently for later epochs. For the first epoch we note that for $t \leq I_0$, $\delta_t = 0$ since SGD performs identical moves in both cases. At $t = I_0 + 1$
$$\delta_{I_0+1} = \|\boldsymbol{w}_{I_0} - \alpha_{I_0}\nabla f(\boldsymbol{w}_{I_0}, z_i) - (\boldsymbol{w}_{I_0}' - \alpha_{I_0}\nabla f(\boldsymbol{w}_{I_0}', z_i'))\| = \alpha_{I_0}\|\nabla f(\boldsymbol{w}_{I_0}, z_i) - \nabla f(\boldsymbol{w}_{I_0}', z_i')\|,$$
where the second equality follows from the fact that $\boldsymbol{w}_{I_0} = \boldsymbol{w}_{I_0}'$ by the definition of $I_0$. Using Lemma 5 we can say that $\delta_{I_0+1} \leq 2\alpha_{I_0}L_S$ almost surely. Notice here we used data-dependent Lipschitz constant $L_S$ which is only defined for points in set $S$ unlike "global" Lipschitz constant. Now,

$$\delta_{I_0+2} \leq \|\boldsymbol{w}_{I_0+1} - \boldsymbol{w}_{I_0+1}'\| + \alpha_{I_0+1}\|\nabla f(\boldsymbol{w}_{I_0+1}, z_i) - \nabla f(\boldsymbol{w}_{I_0+1}', z_i'))\|.$$

Here although the parameter vectors $\boldsymbol{w}_{I_0+1}$ and $\boldsymbol{w}_{I_0+1}'$ are not the same, $z_{\pi_0(I_0+1)}$ and $z_{\pi_0(I_0+1)}'$ are the same by the definition of $I_0$ (assuming that $I_0 \neq m$). Therefore we get that
$$\delta_{I_0+2} \leq \delta_{I_0+1} + \alpha_{I_0+1}K_g\delta_{I_0+1}$$
with probability 1 since, from Lemma 5 we have that $f$ has a "global" smoothness property for the entire set of at most $2T$ parameter vectors that will be encountered during the coupled training of $S$ and $S^i$. Noting that a similar recursion can be applied all the way to the end of the first epoch, i.e. till $t = m$ we get

$$\delta_m \leq 2\alpha_{I_0}L_S \prod_{t=I_0+1}^{m} (1 + \alpha_t K_g) \leq 2\alpha_{I_0}L_S \exp\left\{\sum_{t=I_0+1}^{m} \alpha_t K_g\right\}, \qquad (4)$$

with probability 1. Moving on to the next epoch we note that we can make the argument above till the next point where the two training sequences differ, i.e., till the $m + I_1 + 1$st step. At this point we have,
$$\delta_{m+I_1+1} \leq \delta_{m+I_1} + \alpha_{m+I_1}\|\nabla f(\boldsymbol{w}_{m+I_1}, z_i) - \nabla f(\boldsymbol{w}_{m+I_1}', z_i'))\|.$$

Since neither the parameter vector nor the training points are the same in the second term, we have no option but to use the almost data-dependent Lipschitz constant to say that
$$\delta_{m+I_1+1} \leq \delta_{m+I_1} + \alpha_{m+I_1}2L_S.$$
Since $\alpha_{m+I_1} < \alpha_{I_0}$, observing that our current bound for $\delta_{m+I_1}$ is larger than $\alpha_{m+I_1}2L_S$. Therefore
$$\delta_{m+I_1+1} \leq 2\delta_{m+I_1}.$$

So, we see that in the second and subsequent epochs, for time step $jm + I_j + 1, 1 \leq j < T/m$ we have the bound
$$\delta_{m+I_j+1} \leq 2\delta_{m+I_j},$$
and for all $t > m + I_1, t \neq I_1, \ldots, I_{T/m-1}$ we have, as before, by the smoothness property that
$$\delta_{t+1} \leq \delta_t(1 + \alpha_{t+1}K_g).$$

Therefore, we have that

$$\delta_T \leq 2\alpha_{I_0} L_S(2)^{\frac{T}{m}-1} \exp\left\{\sum_{t=I_0+1}^{T} \alpha_t K_g\right\} \leq \alpha_0 L_S 2^{T/m} \left(\frac{2T}{m}\right)^{\alpha_0 K_g} \frac{m^{\alpha_0 K_g}}{(1+I_0)^{1+\alpha_0 K_g}}. \quad (5)$$

where, in the first inequality for ease of calculation we have retained the terms of the form $(1 + \alpha_{I_j} K_g)$, $2 \leq j < T/m$ in the product on the right although we can ignore them. In the second inequality we have substituted $\alpha_t = \alpha_0/t$ and used the fact that $\log n \leq H_n \leq \log 2n$ for the $n$-th Harmonic number $H_n$.

Finally, in order to compute $E_r[\delta_T]$ we observe that, since $\pi_0$ is uniformly drawn from the set of permutations of $[m]$, $I_0$ is uniformly distributed on $[m]$. Summing up the last term of (5) over $I_0 \in [m]$ and dividing further by $m$ we get the result. $\qquad\square$

## C  NEURAL NETWORKS: CHARACTERIZATION AND PROOFS

In order to prove Theorem 9 we first need to describe a characterization of Neural Networks that allows us to get a better insight into their smoothness properties. We present the characterization in Section C.1 and the proof in Section C.2.

### C.1  A POLYNOMIAL-BASED CHARACTERIZATION NEURAL NETWORKS

#### C.1.1  NEURAL NETWORK TERMINOLOGY

Neural networks provide a family of parametrized functions of the form we have discussed in Section 4. The parameter vector $\boldsymbol{w} \in \mathbb{R}^n$ is applied over a network structure with layers. In this case we specify $\mathcal{Z}$ to be $\mathbb{R}^d \times \mathbb{R}$, i.e., the data points are from $\mathbb{R}^d$ and the label is from $\mathbb{R}$, i.e., the NN has a single output. We will denote the depth of the network by $H$. The layers will be numbered 0 to $H$ with layer 0 being the *input layer*. The number of neurons in layer $i$ will be $k_i$. For this discussion we assume a fully connected network. We will denote by $w_{j,k}^i$ the weight of the edge from the $j$ neuron of the $i$th layer to the $k$th neuron of the $i + 1$st layer. For the NN with parameters $\boldsymbol{w}$ at a point $\boldsymbol{x} \in \mathbb{R}^d$ we will denote the input into the $j$th neuron of the $i$th layer by $\mathsf{in}_{i,j}(\boldsymbol{w}, \boldsymbol{x})$ and its output by $\mathsf{out}_{i,j}(\boldsymbol{w}, \boldsymbol{x})$. These will be equal only if the former is non-negative. Further, we will assume that all neurons in all layers of the network except the input layer and the output layer have ReLU activation applied to them. In case the output of a node is 0 due to ReLU activation we will say the ReLU gate is *closed* otherwise we will say it is *open*. The label output by the network will be $\mathsf{out}_{H,1} = \mathsf{out}(\boldsymbol{w}, \boldsymbol{x})$. For each of exposition we will assume that $\mathsf{out}(\boldsymbol{w}, \boldsymbol{x}) = 1$ if $\mathsf{in}(\boldsymbol{w}, \boldsymbol{x}) > 0$ and 0 otherwise, i.e., there are only two labels in $\mathcal{Y}$.

#### C.1.2  MULTIVARIATE POLYNOMIALS ASSOCIATED WITH A NEURAL NETWORK

Given a set of indeterminates $\boldsymbol{x} = x_1, \ldots, x_l$, let $\mathcal{P}(\boldsymbol{x})$ be the set of multivariate polynomials on $x_1, \ldots, x_l$ with real coefficients. For any polynomial $p(\boldsymbol{x})$, $i_1, \ldots, i_q \in [l]$ and any $\alpha_1, \ldots, \alpha_q \in \mathbb{R}$ for some $q \leq l$, we will denote by $p(\boldsymbol{x})\left\{x_{i_j} = \alpha_j : j \in [q]\right\}$ the polynomial in $\mathcal{P}(\boldsymbol{x} \setminus \{x_{i_1}, \ldots, x_{i_q}\})$ that is obtained by setting all occurences of $x_{i_j}$ to $\alpha_j$ in $p(\boldsymbol{x})$. In particular, $p(\boldsymbol{x})\{x_i=0\}$ is the polynomial $p(\boldsymbol{x})$ with all monomials containing $x_i$ removed, and $p(\boldsymbol{x})\{x_i=1\}$ retains all the monomials of $p(\boldsymbol{x})$ but those monomials that contain $x_i$ appear without the term $x_i$.

Returning to NNs, let us consider two sets of indeterminates: $\boldsymbol{x} = \{x_i : i \in [d]\}$ and $\boldsymbol{w} = \{w_{j,k}^{(i)} : 0 \leq i < H, 1 \leq j \leq k_i, 1 \leq k \leq k_{i+1}\}$ and $k_0 = d$. For a fully connected NN defined in Sec. C.1.1 we will say that it has the following polynomial associated with it:

$$\phi(\boldsymbol{w}, \boldsymbol{x}) = \sum_{j_0=1}^{k_0} \sum_{j_1=1}^{k_1} \cdots \sum_{j_{H-1}=1}^{k_{H-1}} x_{j_0} w_{j_0,j_1}^{(0)} w_{j_1,j_2}^{(1)} \cdots w_{j_{H-1},1}^{(H-1)}.$$

Note that the output layer has only one neuron. We will refer to this as the *base polynomial* of the NN. The base polynomial associated with the $j$th neuron in layer $i$ can be derived from the base polynomial of the network as follow

$$\phi_{i,j}(\boldsymbol{w}, \boldsymbol{x}) = \frac{\phi(\boldsymbol{w}, \boldsymbol{x}) \left\{ w_{l_1,l_2}^{(i)}=0, w_{l_4,l_5}^{(l_3)}=1:l_1 \in [k_i]\setminus\{j\}, l_2 \in [k_{i+1}], l_3 > i, l_4 \in [k_{l_3}], l_5 \in [k_{l_3+1}] \right\}}{\prod_{p=i+1}^{H} k_i}.$$

Due to ReLU activations varying at different points there is no single polynomial that captures the output of the NN everywhere in $\mathbb{R}^n \times \mathbb{R}^d$. However, the following observation shows a way of defining polynomials that describe the output over certain subsets of the space.

**Observation 14.** *Given $\boldsymbol{w} \in \mathbb{R}^n$ and $\boldsymbol{x} \neq (0,\dots,0) \in \mathbb{R}^d$, $i \in [H]$, and $j \in [k_i]$ such that $\mathsf{in}_{l_1,l_2}(\boldsymbol{w},\boldsymbol{x}) \neq 0$ for all $1 \le l_1 \le i$ and all $1 \le l_2 \le k_{l_1}$, there is a $\varepsilon > 0$ depending on $\boldsymbol{w}, \boldsymbol{x}$ such that, for all $\boldsymbol{w}'$ with $\|\boldsymbol{w} - \boldsymbol{w}'\| < \varepsilon$,*
$$\mathsf{in}_{i,j}(\boldsymbol{w}', \boldsymbol{x}) = \mathsf{out}_{i,j}(\boldsymbol{w}', \boldsymbol{x}) = \phi_{i,j}(\boldsymbol{w}', \boldsymbol{x}) \left\{ w_{l_2,l_3}^{(l_1)}=0:l_1<i, l_2 \in [k_{l_1}], l_3 \in [k_{l_2+1}] \text{ and } \mathsf{in}_{l_1,l_2}(\boldsymbol{w},\boldsymbol{x})<0 \right\}.$$

*Proof.* Since $\mathsf{in}_{i,j}(\boldsymbol{w}, \boldsymbol{x})$ is strictly separated from 0 and there are only a finite number of neurons in the network, there must be a $\varepsilon$ small enough for which all open ReLU gates remain open and all closed gates remain closed. It is easy to verify that the description of $\mathsf{out}(\boldsymbol{w}', \boldsymbol{x})$ given above is correct since it simply sets to 0 all the variables corresponding to weights originating in neurons whose ReLU gate is closed. $\square$

## C.2 PROOF OF THEOREM 9

*Proof of Theorem 9.* The idea behind this proof is due to Milne (2019) who used it for a different purpose. From Observation 14 it follows that if we have $\boldsymbol{x} \neq (0,\dots,0) \in \mathbb{R}^d$ such that $\mathsf{in}_{i,j}(\boldsymbol{w}, \boldsymbol{x}) \neq 0$ for all $1 \le i \le H$ and all $1 \le j \le k_i$, then $\mathsf{out}(\boldsymbol{w}, \boldsymbol{x})$ is, in fact, just the polynomial $\phi(\boldsymbol{w}', \boldsymbol{x})$ within a small neighbourhood of $\boldsymbol{w}$. Therefore it is doubly differentiable. Since the loss function is also differentiable, we are done for all such values of $\boldsymbol{x}$.

So now let us consider the set of points $\boldsymbol{x}$ for which $i$ is the smallest layer index such that $\mathsf{in}_{i,j}(\boldsymbol{w}, \boldsymbol{x}) = 0$. In case there are two such indices, we break ties using the neuron index $j$. By Observation 14, in a neighbourhood of $\boldsymbol{w}$, $\mathsf{in}_{i,j}(\boldsymbol{w}, \boldsymbol{x})$ is a polynomial in $\boldsymbol{w}$ and $\boldsymbol{x}$ for each $\boldsymbol{x}$.

Now, we consider two cases. In the first case, $\mathsf{out}_{i-1,j'}(\boldsymbol{w}, \boldsymbol{x}) = 0$ for all $j' \in [k_{i-1}]$, i.e., all the ReLU gates from the previous layers are closed because $\mathsf{in}_{i-1,j'}(\boldsymbol{w}, \boldsymbol{x}) < 0$ for all $j' \in [k_{i-1}]$. In this case $\mathsf{out}(\boldsymbol{w}', \boldsymbol{x}) = 0$ everywhere in the neighbourhood guaranteed by Observation 14 and therefore $\ell(\mathsf{out}(\boldsymbol{w}', \boldsymbol{x}), \mathsf{lab}(\boldsymbol{x}))$ is doubly differentiable in the parameter space at $\boldsymbol{w}$ for all such $\boldsymbol{x}$, where we assume that each data point has a label $\mathsf{lab}(\boldsymbol{x}) \in \{0,1\}$ associated with it. We note that this argument is easily portable to the case of a more general label set $\mathcal{Y}$ with the property described in the statement of Theorem 8 since $\mathsf{in}_{H,1}$ will be 0 everywhere in a small neighbourhood.

In the second case we have some $j' \in [k_{i-1}]$ such that $\mathsf{out}_{i-1,j'}(\boldsymbol{w}, \boldsymbol{x}) > 0$. Let $C_{i,j} \subseteq \mathbb{R}^d$ be those $\boldsymbol{x}$ for which this case holds. $C_{i,j}$ contains the solutions to $\mathsf{in}_{i,j}(\boldsymbol{w}, \boldsymbol{x}) = 0$. Since we are working with a specific value of $\boldsymbol{w}$, this simply becomes a polynomial in $\boldsymbol{x}$. In fact, inspecting the definition of base polynomials we note that when $\boldsymbol{w}$ is fixed $\mathsf{in}_{i,j}(\boldsymbol{w}, \boldsymbol{x})$ is simply a linear combination of $x_1, \dots, x_{\mathbb{R}}^d$. This implies that $C_{i,j}$ is a hyperplane in $\mathbb{R}^d$. We note that this argument can also be made of the output node under the condition on the labelset given in the statement of Theorem 8 because for $\mathsf{in}_{H,1}(\boldsymbol{w}, \boldsymbol{x})$ to give a value that lies on the boundary between two sets with different labels for a given $\boldsymbol{w}$, $\boldsymbol{x}$ must be drawn from a set of Lebesgue measure 0.

Since the network size is finite the set of all possible values of $\boldsymbol{x}$ for which case 2 occurs, i.e., $\bigcup_{i \in [H], j \in [k_i]} C_{i,j}$ is a finite union of hyperplanes in $\mathbb{R}^d$ and therefore a set of Lebesgue measure 0. $\square$

*Proof of Proposition 10.* Let us consider partial derivative w.r.t $w_{\ell_1,\ell_2}^{(i)}$. For this let $I_{l_1,l_2}^{i}$ be the matrix of size $W^i$ such that all $I_{l_1,l_2}^{i}[l_1, l_2] = 1$ and rest all entries are zero. So we can write the partial derivative as,
$$\frac{\partial \phi(\boldsymbol{w}, \boldsymbol{x})}{\partial w_{\ell_1,\ell_2}^{(i)}} = W^H \times W^{H-1} \dots W^{i+1} \times I_{l_1,l_2}^{i} \times W^{i-1} \dots W^1 \times \boldsymbol{x}$$

Although we have scalar values both the side, we take spectra norm ($\|.\|_\sigma$) on both the sides, and upper bound R.H.S by product of individual norms.

$$\left| \frac{\partial \phi(\boldsymbol{w}, \boldsymbol{x})}{\partial w_{\ell_1, \ell_2}^{(i)}} \right| \leq \frac{\prod_{l=1}^{H} \|W^l\|_\sigma}{\|W^i\|_\sigma} \times \|\boldsymbol{x}\|_2$$

Now taking $\|.\|_2$ over gradient vector we get the desired result.

$\square$

## D   REPLY OF REVIEWS

### D.1   REVIEWER 1, UXNU

¿ the authors may need to provide some examples to quantify how weak it is and how much improvement we can get by using this weaker version of stability

The standard definition of stability cannot handle non-linearities. We are giving a much weaker version of stability to widen the usability of stability to more realistic frameworks. Specifically when non-linearities are used neural networks cannot be globally parameter Lipschitz and smooth, so in order to fit this assumption we used a.s. support stability to allow for the absence of smoothness with small probability.

¿ What's the difference between Theorem 7 and the generalization results in [Hardt et al., 2016], if ignoring the difference in terms of the assumptions?

Yes, Theorem 7 and Generalization result of Hardt et al looks similar but this probabilistic approach widens the application of stability, we used the a.s. version with $\eta = 0$ which allowed us to give generalization bound for neural networks with non-linearities like ReLU. Moreover support stability allows for data dependent assumptions over set $S$ and the replaced point $z_i'$. Notice now we have added a more stronger generalization error bound for a.s. support stability in Theorem 4

¿ It seems that the major goal of Section 5 is to (1) verify that Assumptions (Definitions 4 & 5) are satisfied. Then why Section 5.1.1 and Section 5.1.2 are necessary, at least in the main part of this paper?

We thank the reviewer for catching this. We have moved the material from Section 5.1 to Appendix.

¿ As claimed in the beginning of this paper, one potential advantage of the developed theoretical framework is to leverage the information of data distribution, or simply, the support of data distribution. However, I do not see this in the theorem or discussion after the theorems. I believe this should be a key point of the new theory, so the authors may need to thoroughly discuss whether or how considering this point can give better generalization results.

We thank the reviewer for pointing this out. In fact our results had the potential for making a clearer connection with data dependence and we have now established it. The stability bound of Theorem 6 now has a data dependent Lipschitz constant $L_S$ in it. We have added a discussion below Theorem 6 on how this allows us to argue that for "good" unknown distributions we expect our stability bound to be good. Here "good" can either mean that $D$ has low variance or it can mean that it is a distribution for which SGD converges.

¿ The definition of i,j(w,x) needs more explanation, why this definition is useful and how to leverage it should be particularly discussed.

¿ In Observation 10, why the results do not rely on the negativeness or positiveness of ini,j(w' , x)?

¿ Novelty seems good, but it still lacks a thorough comparison with prior works.

### D.2   REVIEWER 2, 8TFR

¿ This is because the paper's results are mostly based on the assumptions on the local Lipschitzness and smoothness of the loss function, and there are very few assumptions on the data distribution which is the key to the good generalization performance of deep convolutional nets on image data

¿ the smoothness coefficient of standard deep neural networks could be extremely large in practice, which makes the generalization bound overly large for actual deep learning experiments.

¿ Theorem 9 is an asymptotic convergence result and does not provide a non-asymptotic generalization error bound that holds for finite training data.

¿ The paper's experiments use a one-layer neural network with ReLU activation (the width of the network is not specified in the text). This setting seems too simple for experimenting the generalization framework, and that would be better to include numerical results for more realistic deep learning problems, e.g. a deep CNN or ResNet applied to a more challenging image dataset.

## D.3 REVIEWER 3, GUYV

¿ The authors are required to clarify why the "a.s." is helpful. Is it designed to match the zero-measure nonlinear points in the activation functions?

He has answered his own question! But we need to discuss further. Discuss example presented for UXNu.

¿ the paper needs a proof sketch in the main text. Based on the sketch, the authors need to clarify the significance and difficulty of the proofs.

¿ the authors are required to give a more detailed discussion on the differences and advantages of the a.s. support stability with the many existing stability measures.

¿ To me, the proofs are quite straightforward: the relationship between the stability and the generalization bound needs merely an additive decomposition on the probability compared with the existing results; the a.s. locally smooth and a.s. locally Lipschitz continuous are straightforward given the nonlinear points in ReLU are of zero measure; and the a.s. support stability of the whole ReLU network comes from (1) the stability of a single layer (linear mapping of the weight matrix + a.s. linear mapping of the ReLU) and (2) composition of the multiple layers. These undermines the quality of this paper. Please correct me in the responses if I was wrong.

Response: We request the reviewer to take a closer look at the proof of Theorem 11. The proof is more complicated than he thinks, it is not, in fact, a composition of stabilities across layers. It is a deeper case analysis. Hopefully this will convince the reviewer that the mathematical content is non-trivial. We also request the reviewer to appreciate that from modifying McDiarmid's theorem to analysing SGD to showing the smoothness of ReLU, we have connected a number of different concepts. Often deep mathematics involves building a big structure out of a large number of small steps.

¿ While the authors give estimates for these constants, the upper bounds grow as a linear function of the number of parameters, and an exponential function on the depth. This implies these results can only apply to neural networks with small complexity. An advantage of stability analysis is that it can imply dimension-independent bounds. However, the results of this paper does not inherit this property.

¿ in the recent analysis generalization bounds for the order $O(\log(m) + 1/m)$ have been developed for $\beta$-uniformly stable algorithms, which is much better than the bounds in Thm 3. While Thm 3 considers a relaxation of uniform stability, the results do not recover the existing bounds if $\eta = 0$.

HP: Our a.s. Support stability can also be directly used in conjunction with Feldman and Vondrak's techniques. We have added a theorem stating this in Sec 3.4 and given a proof outline in the appendix.

¿ The upper bounds of Thm 7 also seem to be suboptimal. For example, if 0=0 we expect the generalization error to be 0. However, in this case, the upper bounds there become O(Lg22T/m/(Kgm))

We have fixed it thankyou for pointing it out.

¿ Furthermore, the step size is of the order O(1/t). The step size decays very fast for which the training errors by SGD decay very slowly. Therefore, it is hard to find a good model with this step size scheme.

¿ If we combine Proposition 12 and Thm 7 together, the generalization bounds would involve O(n K2H), where n is the number of parameters and H is the depth. It seems that this approach does not show advantage over complexity-based approach. For example, size-independent generalization bounds have been derived based on Rademacher complexity analysis, see, e.g., Ref [1,2]. Therefore, it is not quite clear to me the usefulness of the support stability since there are no convincing applications. I would suggest the authors to make a detailed comparison between these bounds.

¿ Section 4: In 3.1 should be in Section 3.1 ¿ Section 4: and and ¿ Thm 9: what is the meaning of ab(x) ¿ Section 5.3: it is possible look ¿ Proof of Thm 7: you define t = —Er[wt - wt']— . According to the proof, it seems that t= wt - wt' ¿ Proof of Thm 3: "the concavity of the absolute value function" should be "the convexity of absolute value function"

