# OpenReview forum: "Generalization error bounds for Neural Networks with ReLU activation"
_ICLR.cc/2023/Conference — Submitted to ICLR 2023_

### Official Review · Reviewer_omdi · 2022-10-13

**Confidence:** 4
**Correctness:** 4
**Technical Novelty And Significance:** 2
**Empirical Novelty And Significance:** 2
**Recommendation:** 5

**Clarity, Quality, Novelty And Reproducibility:**

In Thm 3, the authors develop high-probability generalization bounds for algorithms with support stability. The bounds are of the order $O(\sqrt{m}\beta)+1/\sqrt{m}$. This bound is a bit crude. For example, in the recent analysis generalization bounds for the order $O(\beta\log m+1/\sqrt{m})$ have been developed for $\beta$-uniformly stable algorithms, which is much better than the bounds in Thm 3. While Thm 3 considers a relaxation of uniform stability, the results do not recover the existing bounds if $\eta=0$.

The upper bounds of Thm 7 also seem to be suboptimal. For example, if $\alpha_0=0$ we expect the generalization error to be 0. However, in this case, the upper bounds there become $O(L_g^22^{T/m}/(K_gm))$. Furthermore, the step size is of the order $O(1/t)$. The step size decays very fast for which the training errors by SGD decay very slowly. Therefore, it is hard to find a good model with this step size scheme.

If we combine Proposition 12 and Thm 7 together, the generalization bounds would involve $O(nK^{2H})$, where $n$ is the number of parameters and $H$ is the depth. It seems that this approach does not show advantage over complexity-based approach. For example, size-independent generalization bounds have been derived based on Rademacher complexity analysis, see, e.g., Ref [1,2]. Therefore, it is not quite clear to me the usefulness of the support stability since there are no convincing applications. I would suggest the authors to make a detailed comparison between these bounds.

[1]: Golowich, Noah, Alexander Rakhlin, and Ohad Shamir. "Size-independent sample complexity of neural networks." Conference On Learning Theory, 2018.
[2]: Bartlett, Peter L., Dylan J. Foster, and Matus J. Telgarsky. "Spectrally-normalized margin bounds for neural networks." Advances in neural information processing systems 30 (2017).

Minor comments:

Section 4: In 3.1 should be in Section 3.1

Section 4: and and

Thm 9: what is the meaning of ab(x)

Section 5.3: it is possible look

Proof of Thm 7: you define $\delta_t=\|E_r[w_t-w_t']\|$. According to the proof, it seems that $\delta_t=w_t-w_t'$

Proof of Thm 3: "the concavity of the absolute value function" should be "the convexity of absolute value function"

**Strength And Weaknesses:**

Strength:

The almost sure support stability is a relaxation of the uniform stability, and therefore can be applied to more general problems. The authors apply this stability to derive generalization bounds for neural networks with ReLU activations, which have not been studied by uniform stability.

Weakness:

The generalization bounds depend on the global Lipschitz and global smoothness constants. While the authors give estimates for these constants, the upper bounds grow as a linear function of the number of parameters, and an exponential function on the depth. This implies these results can only apply to neural networks with small complexity. An advantage of stability analysis is that it can imply dimension-independent bounds. However, the results of this paper does not inherit this property.



**Summary Of The Paper:**

This paper studies generalization bounds for neural networks based on the algorithmic stability. To this aim, the paper first introduces a new stability called almost sure support stability, which relaxes the uniform stability by allowing the stability to be violated with some probability. The first result is an exponential bound on generalization gaps for algorithmic satisfying almost (sure) support stability. Then, the paper develops a.e. support stability bounds for SGD with locally Lipschitz/smooth functions. Finally, the authors develop bounds on the global Lipschitz/smoothness constants for neural networks with ReLU activation functions. Experimental results are also provided to verify the effectiveness of the theoretical results.

**Summary Of The Review:**

The paper provides a new stability and apply it to problems with local Lipschitz/smooth problems, e.g., neural networks with activation functions. However, the upper bound is a linear function of the number of parameters and an exponential function of the depth. Therefore, it is not clear to me the advantage of this approach over the complexity-based approach.

---

> ### Author Response · Authors · 2022-11-19
> **Incorporated Feldman-Vondrak style bounds and also gave Lipschitz bounds based on spectral complexity following Bartlett et. al. and Golowich et. al.**
>
> We thank the reviewer for insightful comment. Specific responses are below.
>
> > While the authors give estimates for these constants, the upper bounds grow as a linear function of the number of parameters, and an exponential function on the depth. This implies these results can only apply to neural networks with small complexity. An advantage of stability analysis is that it can imply dimension-independent bounds. However, the results of this paper does not inherit this property.
>
> The reviewer is right. The result we had presented was too loose. Thanks the reviewers comment we went deeper into the issue and have revised Sec 5.1 to include a spectral bound on the Lipschitz constant. This spectral bound can, in some cases, be size independent.
>
> > in the recent analysis generalization bounds for the order $O( \log(m) + 1/m)$ have been developed for $\beta$-uniformly stable algorithms, which is much better than the bounds in Thm $3$. While Thm $3$ considers a relaxation of uniform stability, the results do not recover the existing bounds if $\eta=0$.
>
> We thank the reviewer for pointing this out. Our a.s. support stability can also be directly used in conjunction with Feldman and Vondrak’s techniques. We have added restated all our results in these terms now. We request the reviewer to see the revised Sec 3.3.
>
>
> > The upper bounds of Thm 7 also seem to be suboptimal. For example, if 0=0 we expect the generalization error to be 0. However, in this case, the upper bounds there become O(Lg22T/m/(Kgm))
>
>
> We thank the reviewer for catching this. This was a calculation error on our part. This has been fixed now. Please see the new statement of the theorem (it is now Theorem 6).
>
>
> > Furthermore, the step size is of the order O(1/t). The step size decays very fast for which the training errors by SGD decay very slowly. Therefore, it is hard to find a good model with this step size scheme.
>
> Currently we have no response to this objection. However with the use of Feldman and Vondrak's method we hope we can handle slower step size decay in the future.
>
>
> > If we combine Proposition 12 and Thm 7 together, the generalization bounds would involve O(n K2H), where n is the number of parameters and H is the depth. It seems that this approach does not show advantage over complexity-based approach. For example, size-independent generalization bounds have been derived based on Rademacher complexity analysis, see, e.g., Ref [1,2]. Therefore, it is not quite clear to me the usefulness of the support stability since there are no convincing applications. I would suggest the authors to make a detailed comparison between these bounds.
>
> We thank the reviewer for this pointer. We went deeper into this and changed the bounds to reflect the spectral complexity of the network rather than the size. Please see the revision of Sec 5.1
>
> > Section 4: In 3.1 should be in Section 3.1
> > Section 4: and and
> > Thm 9: what is the meaning of ab(x)
> > Section 5.3: it is possible look
> > Proof of Thm 7: you define t = |Er[wt - wt']| . According to the proof, it seems that t= wt - wt'
> > Proof of Thm 3: "the concavity of the absolute value function" should be "the convexity of absolute value function"
>
> We thank the reviewer for pointing out these kinds of typographical errors. We have fixed all of them.

---

### Official Review · Reviewer_gUyv · 2022-10-23

**Confidence:** 4
**Correctness:** 4
**Technical Novelty And Significance:** 3
**Empirical Novelty And Significance:** Not applicable
**Recommendation:** 5

**Clarity, Quality, Novelty And Reproducibility:**

For clarity and novelty, please refer to point 1 above. This paper is of fair quality in its current form.

**Strength And Weaknesses:**

This paper presents a relatively new notion on the stability. The generalization bound based on the stability is also novel but not seems not difficult. The authors are required to clarify why the “a.s.” is helpful. Is it designed to match the zero-measure nonlinear points in the activation functions? Also, the authors are required to give a more detailed discussion on the differences and advantages of the a.s. support stability with the many existing stability measures.

The paper gives detailed proofs in appendices. However, the paper needs a proof sketch in the main text. Based on the sketch, the authors need to clarify the significance and difficulty of the proofs. To me, the proofs are quite straightforward: the relationship between the stability and the generalization bound needs merely an additive decomposition on the probability compared with the existing results; the a.s. locally smooth and a.s. locally Lipschitz continuous are straightforward given the nonlinear points in ReLU are of zero measure; and the a.s. support stability of the whole ReLU network comes from (1) the stability of a single layer (linear mapping of the weight matrix + a.s. linear mapping of the ReLU) and (2) composition of the multiple layers. These undermines the quality of this paper. Please correct me in the responses if I was wrong.

Overall, the authors are encouraged to address the issues above in the responses and the revised draft.

**Summary Of The Paper:**

This paper designs a new notion “almost sure support stability (a.s. support stability)” to measure the learned model’s stability to the disturbance in the training data. The authors then give a generalization bound based on this stability measure. The authors also show that ReLU network is a.s. locally smooth and a.s. locally Lipschitz continuous, which then imply that that the network is a.s. support stability.

**Summary Of The Review:**

Overall, I recommend "5: marginally below the acceptance threshold." I need the authors to give more details in their responses, and if this paper is accepted, the draft needs a revision.

---

> ### Author Response · Authors · 2022-11-19
> **Tried to increase clarity in the paper**
>
> We thank the reviewer for insightful comments. Specific answers are below.
>
> > The authors are required to clarify why the “a.s.” is helpful. Is it designed to match the zero-measure nonlinear points in the activation functions?
>
> Yes, the reviewer is right. The main reason is to provide a theoretically rigorous way of analyzing non-linearities. We would like to point out that other non-linearities like max-pool etc can also be managed in this way.
>
> > the paper needs a proof sketch in the main text. Based on the sketch, the authors need to clarify the significance and difficulty of the proofs.
>
> The reviewer is right. Since there are many theorems it is important to keep the reader oriented. Accordingly we have added several proof outlines, especially of the key results Theorems 2, 6 and 9.
>
> > the authors are required to give a more detailed discussion on the differences and advantages of the a.s. support stability with the many existing stability measures.
>
> Our major competitor is uniform stability and we have discussed above that it is incapable of handling non-linearities like ReLU. Our a.s. support stability is data dependent, but, unlike other data dependent notions like those given by Kuzborskij (2018) it is capable of handling non-linearities rigorously. We request the reviewer to read the last paragraph of Section 4 for an in-depth discussion.
>
> > To me, the proofs are quite straightforward: the relationship between the stability and the generalization bound needs merely an additive decomposition on the probability compared with the existing results; the a.s. locally smooth and a.s. locally Lipschitz continuous are straightforward given the nonlinear points in ReLU are of zero measure; and the a.s. support stability of the whole ReLU network comes from (1) the stability of a single layer (linear mapping of the weight matrix + a.s. linear mapping of the ReLU) and (2) composition of the multiple layers. These undermines the quality of this paper. Please correct me in the responses if I was wrong.
>
> We request the reviewer to take a closer look at the proof of Theorem 11 (it is now Theorem 9 in the redraft). The proof is more complicated than he thinks, it is not, in fact, a composition of stabilities across layers. It is a deeper case analysis. Hopefully this will convince the reviewer that the mathematical content is non-trivial. We also request the reviewer to appreciate that from modifying McDiarmid’s theorem to analysing SGD to showing the smoothness of ReLU, we have connected a number of different concepts. Often deep mathematics involves building a big structure out of a large number of small steps.

---

### Official Review · Reviewer_8Tfr · 2022-10-25

**Confidence:** 3
**Correctness:** 4
**Technical Novelty And Significance:** 2
**Empirical Novelty And Significance:** 2
**Recommendation:** 6

**Clarity, Quality, Novelty And Reproducibility:**

The paper is clearly written and the theoretical analysis has sufficient novelty.

**Strength And Weaknesses:**

Strengths:

1) The paper is well written, and the presentation is clear and satisfactory.

2) The proposed theoretical framework seems useful for proving generalization error bounds for neural nets with non-smooth activation functions such as ReLU.

Weaknesses:

1) I do not find the paper's results adequate to answer the introduction question: "Is there a reasonable definition of stability that incorporates distribution properties and leads to small generalization error?" This is because the paper's results are mostly based on the assumptions on the local Lipschitzness and smoothness of the loss function, and there are very few assumptions on the data distribution which is the key to the good generalization performance of deep convolutional nets on image data. In addition, the smoothness coefficient of standard deep neural networks could be extremely large in practice, which makes the generalization bound overly large for actual deep learning experiments.

2) Theorem 9 is an asymptotic convergence result and does not provide a non-asymptotic generalization error bound that holds for finite training data.

3) The paper's experiments use a one-layer neural network with ReLU activation (the width of the network is not specified in the text). This setting seems too simple for experimenting the generalization framework, and that would be better to include numerical results for more realistic deep learning problems, e.g. a deep CNN or ResNet applied to a more challenging image dataset.

**Summary Of The Paper:**

This paper attempts to prove algorithmic stability-based generalization error bounds for neural nets with ReLU activation functions. To do this, the modified definition of "Almost (Sure) Support Stability" is proposed, which weakens the standard uniform stability definition and requires it to hold with a certain probability over the support set of real data. Theorem 3 shows that a previously-known stability-based generalization bound can be modified using the new definition. Next, the paper defines almost locally Lipschitz and smooth functions, and Theorem 7 shows that assuming a bounded almost locally Lipschitz and smooth coefficients, a generalization guarantee will hold for the SGD optimizer where the number of epochs can be as large as $O(\log(m))$ with $m$ being the number of training samples. Finally, Theorem 9 uses the notions of almost locally Lipschitz and smooth functions to prove an asymptotic convergence guarantee for ReLU-based neural nets.

**Summary Of The Review:**

This paper aims to extend the algorithmic stability generalization analysis to ReLU-based neural networks by defining and analyzing the notion of almost (sure) support stability. While the paper is well written and the analysis takes some steps toward the goal of understanding the generalization of ReLU-based networks, I still think the paper's results are not sufficient to properly bound the generalization error of standard deep neural networks in practice.

---

> ### Author Response · Authors · 2022-11-19
> **We have tried to make the data dependence of our bounds clearer**
>
> We thank the reviewer for insightful comments. Specific answers are below.
>
> > This is because the paper's results are mostly based on the assumptions on the local Lipschitzness and smoothness of the loss function, and there are very few assumptions on the data distribution which is the key to the good generalization performance of deep convolutional nets on image data
>
> We think the reviewer for pointing this out. In fact our results had the potential for making a clearer connection with data dependence and we have now established it. The stability bound of Theorem 6 now has a data dependent Lipschitz constant $L_S$ in it. We have added a discussion below Theorem 6 on how this allows us to argue that for good unknown distributions we expect our stability bound to be good. Here good can either mean that $D$ has low variance or it can mean that it is a distribution for which SGD converges.
>
>
> > the smoothness coefficient of standard deep neural networks could be extremely large in practice, which makes the generalization bound overly large for actual deep learning experiments.
>
> The reviewer is right. This is an issue. However if the spectral complexity of the NN is low enough the generalization bound can work. We request the reviewer to read the revised discussion in Sec 5.1.
>
> > Theorem 9 is an asymptotic convergence result and does not provide a non-asymptotic generalization error bound that holds for finite training data.
>
> Theorem 9 is now Theorem 6 in the new presentation. We have taken care to give the proper non-asymptotic value in this and subsequent results so that it is clear what the rates of decay are. We realize that this is important in the context of generalization error and thank the reviewer for pointing out this omission.
>
> > The paper's experiments use a one-layer neural network with ReLU activation (the width of the network is not specified in the text). This setting seems too simple for experimenting the generalization framework, and that would be better to include numerical results for more realistic deep learning problems, e.g. a deep CNN or ResNet applied to a more challenging image dataset.
>
> We have not been able to do more experiments in the rebuttal period since we were working on fixing the theoretical aspects. Since the reviewer feels this is important we will do these experiments and present them in future discussion.

---

> > ### Comment · Reviewer_8Tfr · 2022-11-27
> > **Thanks for the response**
> >
> > I thank the authors for their thoughtful responses. I think the response addresses my comments 1 and 2 satisfactorily and I raise my score to 6 accordingly.

---

### Official Review · Reviewer_UXnu · 2022-10-25

**Confidence:** 4
**Correctness:** 4
**Technical Novelty And Significance:** 2
**Empirical Novelty And Significance:** 2
**Recommendation:** 5

**Clarity, Quality, Novelty And Reproducibility:**

The clarity needs to be improved. Many notations are confusing and the authors do not clearly illustrate why they are needed and how they are used in proving the main results.

Novelty seems good, but it still lacks a thorough comparison with prior works.


**Strength And Weaknesses:**

Strength:
* This paper provides a new notion of algorithmic stability, which is weaker than prior works and can lead to a meaningful generalization error bound.
* This paper proves the generalization of SGD under a new set of assumptions on the training objectives including “almost locally Lipschitz” and “almost locally smooth”.
* This paper further establishes the generalization of SGD for training NN with ReLU activations by showing that the objective function satisfies the almost locally Lipschitz/smooth conditions with certain parameters.

Weakness:
* I can understand that involving the almost sure definition and considering the support of $D$ can certainly weaken the original definition of uniform stability. However, the authors may need to provide some examples to quantify how weak it is and how much improvement we can get by using this weaker version of stability.
* What’s the difference between Theorem 7 and the generalization results in [Hardt et al., 2016], if ignoring the difference in terms of the assumptions?
* The neural network terminology in Section 5.1 is confusing. It seems that the major goal of Section 5 is to (1) verify that Assumptions (Definitions 4 & 5) are satisfied. Then why Section 5.1.1 and Section 5.1.2 are necessary, at least in the main part of this paper?
* As claimed in the beginning of this paper, one potential advantage of the developed theoretical framework is to leverage the information of data distribution, or simply, the support of data distribution. However, I do not see this in the theorem or discussion after the theorems. I believe this should be a key point of the new theory, so the authors may need to thoroughly discuss whether or how considering this point can give better generalization results.
* The definition of $\phi_{i,j}(w,x)$ needs more explanation, why this definition is useful and how to leverage it should be particularly discussed.
* In Observation 10, why the results do not rely on the negativeness or positiveness of $in_{i,j}(w’,x)$?


**Summary Of The Paper:**

This paper aims to establish a sharp generalization error bound for NN with ReLU activation when the network size doesn’t grow with the training set size. This paper provides a weakened version of uniform stability and proves the generalization guarantees of SGD for training over locally Lipschitz and smooth objectives. Moreover, the authors apply the developed generalization analysis to prove the generalization bound of SGD for training NN with ReLU activations, which yields a vanishing generalization bound when the NN size doesn’t grow with the training sample size.

**Summary Of The Review:**

Overall this paper gives some new notions and analyses for studying the generalization of SGD. However, I am not fully convinced by the novelty and contribution of this paper as some discussions and comparisons to prior works are missing.

---

> ### Author Response · Authors · 2022-11-19
> **We made the data dependent nature of our bound clearer.**
>
> We thank the reviewer for the insightful comments. Specific questions are answered below.
>
> > the authors may need to provide some examples to quantify how weak it is and how much improvement we can get by using this weaker version of stability
>
> The standard definition of stability cannot handle non-linearities. We are giving a much weaker version of stability to widen the usability of stability to more realistic frameworks. Specifically when non-linearities are used neural networks cannot be globally parameter Lipschitz and smooth, so in order to fit this assumption we used a.s. support stability to allow for the absence of smoothness with small probability.
>
>
>
> > What’s the difference between Theorem 7 and the generalization results in [Hardt et al., 2016], if ignoring the difference in terms of the assumptions?
>
> Yes, Theorem 7 and Generalization result of Hardt et al looks similar but this probabilistic approach widens the application of stability, we used the a.s. version with $\eta = 0$ which allowed us to give generalization bound for neural networks with non-linearities like ReLU. Moreover support stability allows for data dependent assumptions over set $S$ and the replaced point $z^{'}_i$. Notice now we have added a more stronger generalization error bound for a.s. support stability in Theorem 4
>
>
> > It seems that the major goal of Section 5 is to (1) verify that Assumptions (Definitions 4 $\&$ 5) are satisfied. Then why Section 5.1.1 and Section 5.1.2 are necessary, at least in the main part of this paper?
>
> We thank the reviewer for catching this. We have moved the material from Section 5.1 to Appendix.
>
> > As claimed in the beginning of this paper, one potential advantage of the developed theoretical framework is to leverage the information of data distribution, or simply, the support of data distribution. However, I do not see this in the theorem or discussion after the theorems. I believe this should be a key point of the new theory, so the authors may need to thoroughly discuss whether or how considering this point can give better generalization results.
>
> We thank the reviewer for pointing this out. In fact our results had the potential for making a clearer connection with data dependence and we have now established it. The stability bound of Theorem 6 now has a data dependent Lipschitz constant $L_S$ in it. We have added a discussion below Theorem 6 on how this allows us to argue that for good unknown distributions we expect our stability bound to be good. Here good can either mean that $D$ has low variance or it can mean that it is a distribution for which SGD converges.
>
>
> > The definition of $\phi_{i,j}(w,x)$ needs more explanation, why this definition is useful and how to leverage it should be particularly discussed.
>
> This material has now been moved to the appendix. In case the paper is accepted and we have to include it in the supplementary material we will explain it further.
>
>
> > In Observation 10, why the results do not rely on the negativeness or positiveness of $\mbox{in}_{i,j}(w' , x)$?
>
> The key point is that the discontinuity occurs only if $\mbox{in}_{i,j}(w' , x) = 0$. If it is negative then within a small epsilon it remains negative and so the ReLU gate remains closed. If it is positive then within a small epsilon it remains positive and so the ReLU gate remains open.
>
> > Novelty seems good, but it still lacks a thorough comparison with prior works.
>
> We have added references to some more papers, especially Bartlett et. al. (2017) and Golowich et. al. (2018)

---

### Decision · Program_Chairs · 2023-01-20

**Decision:**

Reject

**Justification For Why Not Higher Score:**

After the authors revised the paper, the novelty of the paper became unclear because they borrowed several results from existing work and combined them with their work. It is also not thoroughly discussed how and in what sense the derived bound is tight against existing work. To fix this issue, this paper would require a major revision so that the organization becomes more consistent. Moreover, some results are still restrictive. For these reasons, this paper is not yet matured to be accepted by ICLR.

**Justification For Why Not Lower Score:**

N/A

**Metareview: Summary, Strengths And Weaknesses:**

This paper gives a new generalization error bound of ReLU neural network by utilizing the notion of almost sure (a.s.) support stability. Thanks to the technique, we can potentially give a tighter bound because the a.s. support stability requires weaker condition.

Obtaining a new generalization bound with a new technique is interesting and important.
However, this paper fails to derive such a bound in a solid and convincing way. The bound originally derived by the authors was not tight and they improved some of the bounds according to a reviewer's comment. It is nice, but the overall organization of the paper turned out to be patchy. I think the paper should be re-written carefully to make the paper more consistent. Moreover, it is not clearly discussed how their bound improves compared with the existing bounds.
Another concern is that the stability bound for SGD is given only for the step size of O(1/t) which is too small. It is expected that this result is generalized to more general step size setting or the authors give convincing argument to support considering only O(1/t) step size.